# Indirect protection from vaccinating children against influenza in households

Tim K. Tsang[1,2], Vicky J. Fang[1], Dennis K.M. Ip[1], Ranawaka A.P.M. Perera [1,3], Hau Chi So[1], Gabriel M. Leung[1], J.S.Malik Peiris[1,3], Benjamin J. Cowling [1] & Simon Cauchemez[4,5,6]

Vaccination is an important intervention to prevent influenza virus infection, but indirect protection of household members of vaccinees is not fully known. Here, we analyze a cluster household randomized controlled trial, with one child in each household randomized to receive influenza vaccine or placebo, for an influenza B epidemic in Hong Kong. We apply statistical models to estimate household transmission dynamics and quantify the direct and indirect protection of vaccination. Direct vaccine efficacy was 71%. The infection probability of unvaccinated household members in vaccinated households was only 5% lower than in control households, because only 10% of infections are attributed to household transmission. Even when that proportion rises to 30% and all children are vaccinated, we predict that the infection probability for unvaccinated household members would only be reduced by 20%. This suggests that benefits of individual vaccination remain important even when other household members are vaccinated.

[1] WHO Collaborating Centre for Infectious Disease Epidemiology and Control, School of Public Health, Li Ka Shing Faculty of Medicine, The University of Hong Kong, Hong Kong, China. [2] Department of Biostatistics, College of Public Health and Health Professions, University of Florida, Gainesville, FL 32610, USA. [3] Centre of Influenza Research, Li Ka Shing Faculty of Medicine, The University of Hong Kong, Hong Kong, China. [4] Mathematical Modelling of Infectious Diseases Unit, Institut Pasteur, 25-28 Rue du Docteur Roux, 75015 Paris, France. [5] Center of Bioinformatics, Biostatistics and Integrative Biology, Institut Pasteur, 25-28 Rue du Docteur Roux, Paris 75015, France. [6] CNRS UMR2000: Génomique évolutive, modélisation et santé, Institut Pasteur, 25-28 Rue du Docteur Roux, Paris 75015, France. These authors contributed equally: Benjamin J. Cowling and Simon Cauchemez.  Correspondence and requests for materials should be addressed to B.J.C.  (email: bcowling@hku.hk)

Influenza causes substantial morbidity and mortality in humans every year[1,2]. Children generally face the highest risk of influenza virus infection each year[3,4], while the risk of more severe disease for infected people is highest at the extremes of age[5,6]. Most transmission is thought to occur in indoor settings including households, schools, and workplaces[7–9].

Vaccination is one of the most important tools to prevent infection and transmission of influenza viruses. While the direct effect of vaccination has been demonstrated in both vaccine trials and community studies[10,11], the conditions under which indirect protection of vaccination starts to become significant are less clear[12]. Vaccination campaigns targeting a large fraction of children can substantially increase herd immunity[13–15] but the impact is less clear when the intervention is performed at a smaller scale. For example, can we expect parents and siblings of a child to benefit from that child's vaccination, even when vaccination coverage remains limited at the population level? The evidence here is more mixed with two small household studies reporting indirect protection against influenza-like illness[16,17], while a third one reported no significant indirect protection against influenza virus infection[18].

The absence of indirect protection at the household level would seem paradoxical since (i) children are important vectors of influenza transmission, (ii) a substantial proportion of transmission events are believed to happen in the household

environment[19,20], and (iii) the vaccine has a strong direct effect. The lack of clear demonstrated effect could also be due to small sample sizes and/or limitations of the statistical analyses used in past studies.

Here, to assess the potential indirect benefits of influenza vaccination in the household environment, we analyzed data from a randomized placebo-controlled trial of influenza vaccination in children[21] with sophisticated statistical and mathematical models that provide a thorough characterization of the dynamics of influenza transmission in households and the impact of vaccination on these dynamics. We estimate that the indirect protection is limited (~20% reduction of risk), which is lower than the direct vaccine efficacy (71% reduction of risk). This suggests that the benefits of individual vaccination remain important even when other household members are vaccinated.

## Results

**Study participants.** Seven hundred and ninety-six households were enrolled in the study, including 796 children aged 5–17 who were randomly allocated to receive influenza vaccination or placebo and their 2234 household contacts (i.e., other members of the household). Of those 2234 contacts, 225 (10%) reported receiving vaccination during the study period. Serologic data from these contacts were not used in the analysis because of the difficulty in interpreting serology following vaccination[22]. In

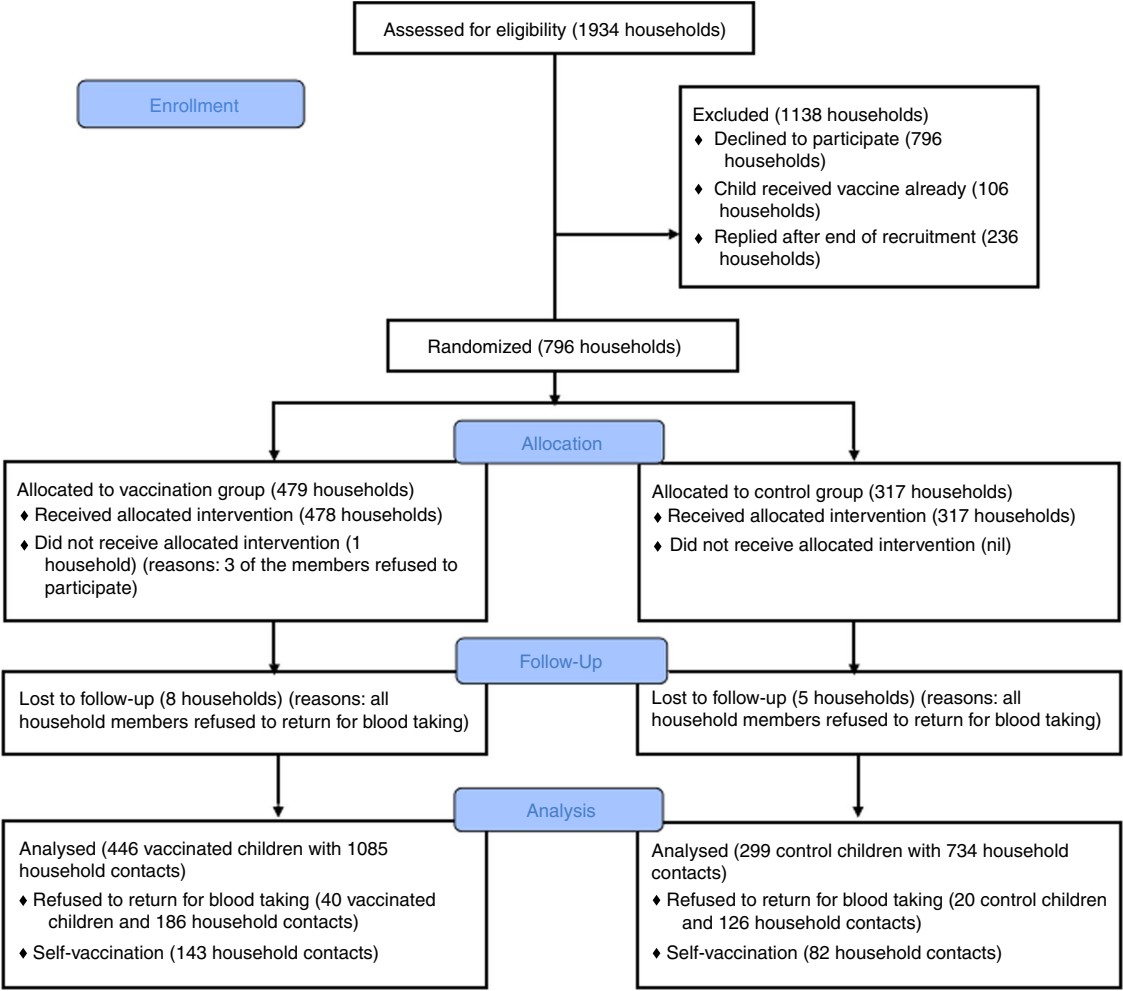

**Fig. 1** Flow chart of participants in our study. Since our inferential framework is able to impute infection status, some of the individuals with missing final outcome could nonetheless be included in the analysis. We include households with at least one household members with paired sera that covered the epidemic

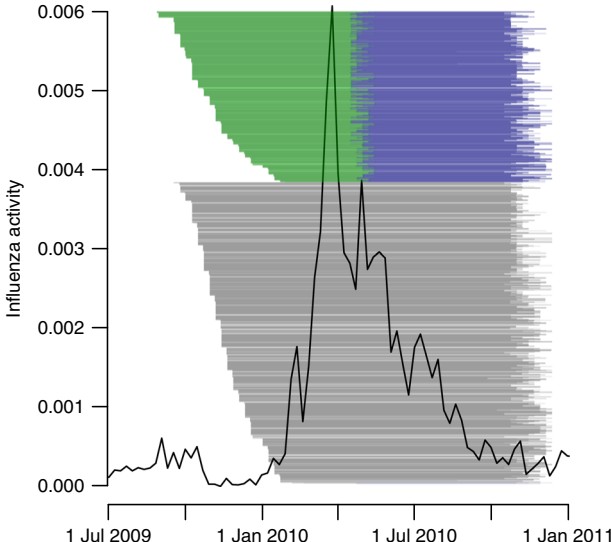

**Fig. 2** Timeline of the study and influenza virus activity for influenza B epidemic in Hong Kong. The black line denotes the local influenza activity in Hong Kong, as approximated by influenza-like illness consultation rates multiplied by the proportion of laboratory specimens testing positive for influenza B virus (ILI + proxy). The green, blue, and black lines indicate the pairs of sera drawn in rounds 1 + 2, rounds 2 + 3, and rounds 1 + 3, respectively

addition, 312 household contacts did not provide serum samples. For the modeling study, we analyzed data from 745 households (2564 participants) in which at least one household member had paired sera that covered the epidemic. Of the 745 households, 446 and 299 households included a child that was randomized to receive influenza vaccine or placebo, respectively (Fig. 1).

There were three rounds of sera collection in our study (Fig. 2), and the characteristics of household contacts in the vaccine and control groups were similar (Supplementary Tables 1–3). The surveillance data in Hong Kong indicated that almost all of the paired sera could cover the epidemic (Fig. 2). In total, we collected 3185 paired sera. From these sera, we identified 161 influenza B infections, defined as 4-fold or greater rises in antibody titers against B/Brisbane/60/2008-like virus measured by the hemagglutination inhibition assay for at least one paired serum. The proportion of child contacts in control households that had 4-fold or greater rise was 16% (20/128), which were higher than 9% (17/194) in the child contacts of vaccine recipients. However, those differences were not statistically significant (Supplementary Table 4).

**Direct effect of vaccination.** We found that children who had received influenza vaccination had a lower susceptibility compared with children who had received the placebo (relative susceptibility: 0.29; 95% CI: 0.17, 0.47, Fig. 3a). Models assuming no direct effect of vaccination performed substantially worse (ΔDIC: 32.4). The vaccine efficacy, computed by one minus relative susceptibility, was therefore 71% (95% CI: 53%, 83%).

**Household transmission dynamics.** Based on the data from both vaccinated and control households, we estimated that unvaccinated adult contacts had lower susceptibility than unvaccinated child contacts (relative susceptibility: 0.39; 95% CI: 0.28, 0.54, Fig. 3b). Ignoring this difference substantially worsened model fit (ΔDIC: 33.0). We also estimated that the relative susceptibility of unvaccinated contacts with an intermediate level of HAI titer and with a high level of HAI titer was 0.48 (95% CI: 0.23, 0.90) and

0.42 (95% CI: 0.17, 0.89), respectively, compared with those with a low level of HAI titer, respectively (Fig. 3c). The model without protection effect from pre-season HAI titers performed substantially worse (ΔDIC: 14.0).

We estimated the probability of infection from the community over the study period for children and adults with a low level of HAI titer was 12% (95% CI: 10%, 15%) and 5% (95% CI: 4%, 6%), respectively (Fig. 4a). When exposed to an infected member in a household of size 2 or 3, children and adults with low levels of HAI titers had a probability of infection of 18% (95% CI: 7%, 34%) and 8% (95% CI: 3%, 14%), respectively (Fig. 4b), while those in a household of size larger than or equal to 4 had a probability of infection of 7% (95% CI: 2%, 14%) and 3% (95% CI: 1%, 6%), respectively. Models ignoring the effect of household size performed substantially worse (ΔDIC: 9.3).

For this influenza B epidemic, we estimated that the proportion of cases attributed to household transmission for all households, vaccine households and control households was 12% (9%, 21%), 10% (6%, 21%), and 13% (9%, 24%), respectively.

**Indirect effect of vaccination.** We evaluated the effect of two vaccination strategies (strategy 1: vaccinate one child in each household, i.e., the study design of our trial; strategy 2: vaccinate all children in the household) on the probability of infection for unvaccinated contacts by simulation (Supplementary Figure 1). We found that, compared to the no vaccination scenario, the probability of household infection for unvaccinated adult contacts was almost halved under both strategies, with a relative probability of 0.69 (95% posterior predictive interval (PPI): 0.54, 0.85) under strategy 1 and 0.55 (95% PPI: 0.40, 0.74) under strategy 2 (Fig. 5). However, the reduction to the total probability of infection was only marginal (relative probability 0.95; 95% PPI: 0.90, 0.99 for strategy 1 and 0.93; 95% PPI: 0.87, 0.98 for strategy 2) because community was by far the main source of infection in our study (Supplementary Figure 1).

We estimated that, in this influenza B epidemic, only 10% of cases were attributed to household transmission (hereafter we denoted this proportion as $P$); but other studies estimated this proportion at around 30%[19,20] in influenza A outbreaks. We therefore assessed how estimates of the indirect benefit of vaccination would be impacted if proportion $P$ increased, for a constant probability of infection from the community. We found that the proportion $P$ had little impact on the reduction of household transmission due to strategies 1 and 2 (Fig. 5). Indeed, for adult contacts, irrespective of $P$, the relative probability of infection from a household member was around 0.68 in strategy 1 and 0.53 in strategy 2. However, as $P$ increased, the indirect effect of vaccination on the total probability of infection became more important. Overall, when $P$ was equal to 30%, the relative probability of infection was 0.89 (95% PPI: 0.80, 0.96) and 0.83 (95% PPI: 0.72, 0.93) for strategy 1 and 2, respectively, compared with no vaccination (Fig. 5). Sensitivity analyses suggested that these results were robust with respect to the small proportion of infection with inconclusive source (Supplementary Figures 2–5).

**Model variations.** Models assuming difference in infectivity between children and adults performed substantially worse (ΔDIC: 32.0). We explored models that incorporated potential differences in infectivity between vaccinated and non-vaccinated individuals but the sample size was insufficient to provide robust estimates.

**Model adequacy.** We simulated 1000 data sets with parameter values drawn from the posterior distribution. The predicted final

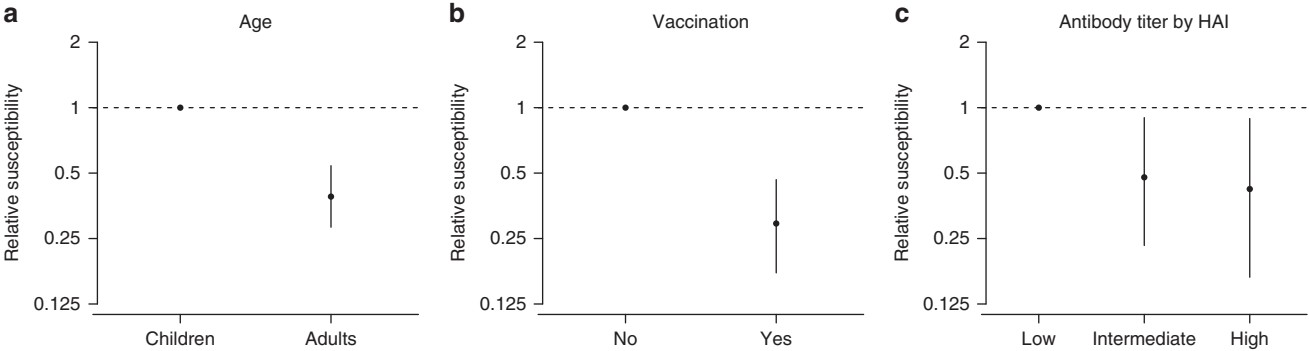

**Fig. 3** Factors affecting the probability of infection. Point and line indicates the point estimate and the 95% credible interval of the relative susceptibility, respectively. 95% credible intervals are constructed by using MCMC to fit the data with digraph model. **a** Estimated relative susceptibility of age. **b** Estimated relative susceptibility of vaccinees. **c** Estimated susceptibility of contacts with intermediate (20 or 40) and high ( > 40) level of pre-season antibody titers, compared with contacts with low ( < 20) level of pre-season antibody titers measured by the hemagglutination inhibition assay, respectively

size distribution was consistent with the observed data and the model fit was judged adequate (Supplementary Table 5).

**Inference and model validation**. In the simulation study, we found no important systematic bias. Moreover, 86 to 100% (depending on the parameter) of the 95% credible intervals covered the simulation value in the simulation study (out of 50 simulated data sets), suggesting that the algorithm was able to estimate adequately the posterior distribution (Supplementary Table 6).

## Discussion

In this study, we estimated the transmission dynamics of influenza B virus in households, exploring factors affecting transmission and quantifying the effectiveness of direct protection for children and its indirect benefit for their household contacts. The direct protection was estimated here by serology to be 71% and this was consistent with previous estimates of vaccine efficacy based on PCR-confirmed infections of 66% (95% confidence interval: 31–83%)[21]. We also tested the hypothesis that household members can benefit indirectly from vaccination of other household members due to prevention of introduction of influenza viruses into households.

We found that vaccination could reduce the probability of household transmission. However, its impact on the overall probability of infection in household contacts was small because, in this influenza B epidemic, household transmission represented only about 10% of all transmission events. The estimated proportion of transmission occurring in households was surprisingly low given other studies found it to be closer to 30%[19,20] for influenza A epidemics. Therefore, we conducted further simulations to evaluate the indirect benefits of vaccination for household members when the proportion of household transmissions was higher. We estimated that when a third of transmissions occurred in households, the probability of infection of adults could be reduced by 20% if all children were vaccinated in the household. To be able to assess when indirect benefits of vaccination can be expected in the household setting, it is therefore important to better understand factors that may drive variations in the relative contribution of households to the overall epidemic. These variations might be due to differences in the influenza strains (our study was based on influenza B infections, while the previous estimates were mostly based on influenza A infections). It is also possible that the probability of infection from the community

might be stronger in Hong Kong due for example to crowded public transportation system and schools.

If a substantial proportion of individuals was vaccinated in the community, the probability of infection from the community could decrease due to herd immunity as shown in other studies[13–15]. In our study, 48/534 (9%) children and 177/1700 (10%) adults were vaccinated. However, this may not reflect the population coverage as our study selected household with at least one unvaccinated child. From a separated household transmission study conducted in Hong Kong[23,24], 39/218 (18%) of child contacts and 111/923 (12%) of adult contacts of index cases were vaccinated. Another household study reported that the vaccine coverage for elderly was 27%[25]. Therefore, the overall vaccine coverage rate in Hong Kong is low and the results of our study cannot directly be compared with such studies.

In the presence of a household member infected by influenza B/Victoria, we estimated that the transmission probability was 14% for children and 5% for adults, with a pre-season titer of < 20. These estimates were similar to those from a case-ascertained study of influenza B/Victoria virus transmission conducted in Hong Kong[26]. Moreover, those estimates were also generally similar to estimates of the secondary infection probabilities for influenza A viruses[23,27,28]. Our study estimated that children were around four times more susceptible to influenza B/Victoria virus infection than adults. This is consistent with a separate case-ascertained study[26] and the age distribution of influenza B/Victoria cases in surveillance data[29,30]; but higher than estimates obtained for influenza A virus that indicate children are about twice as susceptible as adults[19,27,28,31]. Given that our estimate was adjusted for pre-season HAI titer, and the impact of behaviors on transmission should be similar for influenza A and B viruses, these differences may be related to inherent differences in the transmissibility of influenza A and B viruses[32].

We found that an HAI titer of 20–40 was associated with 52% protection, compared with an HAI titer of < 20. This was similar to the previous estimates of 50% protection associated with an HAI titer of 40, compared with an HAI titer of < 10[33]. We found that the person-to-person probability of transmission in households with smaller number of household members was higher, in agreement with other studies[19,34–36].

Our study has some limitations. First, influenza virus infections in our study were identified by examining consecutive HAI titers for 4-fold or greater rises, which can suffer measurement error for various reasons[37]. We decided to use serology instead of PCR because PCR was unable to detect asymptomatic infection since

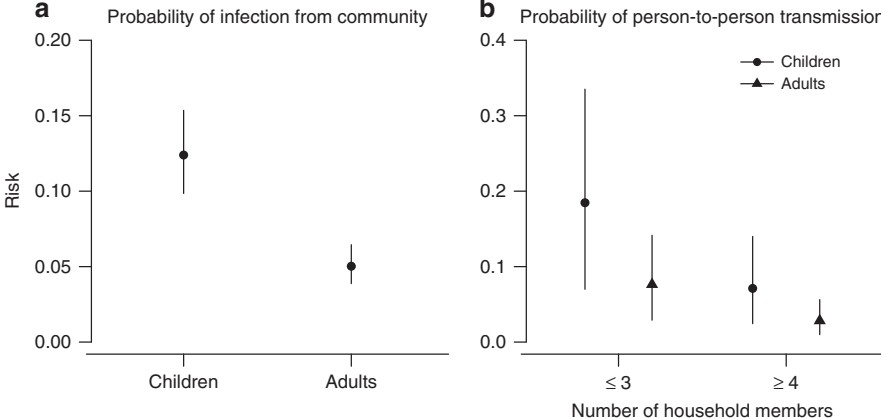

**Fig. 4** Estimated probability of infection from the community and person-to-person transmission in households for individuals with a lower level of titer. Point and line indicates the point estimate and the 95% credible interval of the estimate, respectively. 95% credible intervals are constructed by using MCMC to fit the data with digraph model. **a** Estimated probability of infection from the community over the study period. **b** Estimated probability of person-to-person transmission in households, as a function of household size

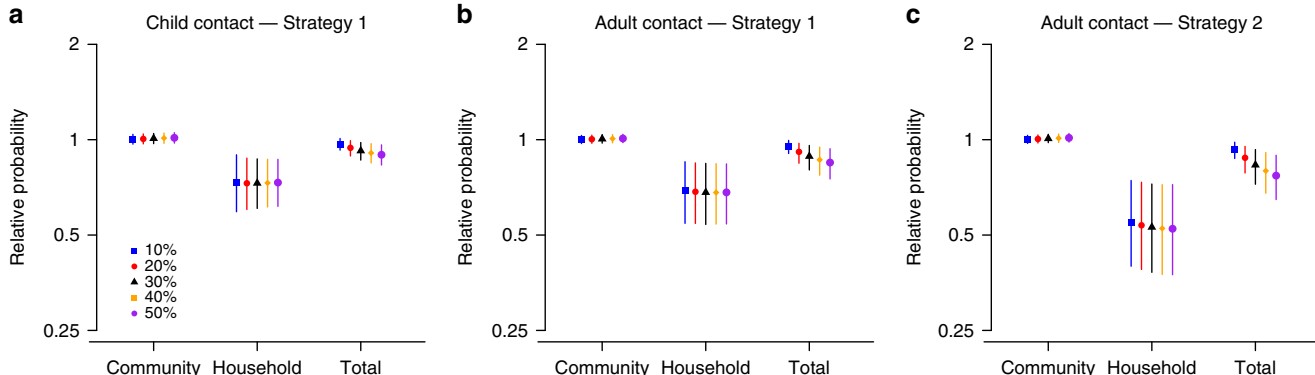

**Fig. 5** The relative probability of infection for household contacts of vaccinated children, under two vaccination strategies compared with no vaccination strategy. The two strategies are 1: vaccination one child per household and strategy 2: vaccinating all children in the household. The relative probabilities are presented for infections from the community only (abbreviated as "Community" in x-axis), infections from household members only ("Household"), and all infections ("Total"). Results are presented for different assumptions about the proportion P of cases attributed to household transmission: 10% (blue), 20% (red), 30% (black), 40% (orange), and 50% (purple). Results for vaccination strategy 1 are presented in (**a** and **b**); those for strategy 2 in (**c**). 95% posterior predictive intervals are constructed with 10,000 simulated epidemics based on the estimated posterior distribution of model parameters (Supplementary Methods)

swabs were only collected when the participants reported symptoms. Second, infections for vaccinated children may be missed because of the ceiling effect of HAI titers[38]. For example, some vaccinated children may have a very high post-vaccination HAI titer so that even if they were infected after vaccination, their titer would not increase by 4-fold. This phenomenon could lead to the overestimation of the direct benefits of vaccination. However, the estimate of vaccine efficacy based on HAI titers here was very similar to that based on PCR-confirmed influenza[21]. Moreover, only 5% (23/467) of vaccinees had a post-vaccination HAI titers of > 640, while our ceiling for HAI titers was 2560. Third, we assumed that infection processes in the different households were independent of each other in our analysis. This seems to be a reasonable assumption because households participating in the study were just a small proportion of those living in Hong Kong, and around 10–30% of people in Hong Kong received influenza vaccination. Finally, each household may have slightly different time of sera collection in each round due to logistical reasons while our analysis assumed that each household shared the probability of infection from the community. However, most of the paired sera covered the epidemic period, based on influenza surveillance in Hong Kong.

In conclusion, we used serology from a household cohort study to infer household transmission dynamics and evaluate the direct and indirect benefits at the household level of vaccinating one child per household. We showed that the indirect benefits depended on the probability of household transmission. We found that in a reasonably optimistic scenario where a third of transmissions occurred in households vaccinating all children in a household provided limited indirect protection (~20%), which is lower than the direct vaccine efficacy (71%). This suggests that the benefits of individual vaccination remain important even when other household members are vaccinated.

## Methods

**Study design**. Data were collected in a community-based randomized controlled trial (ClinicalTrials.gov NCT00792051) aiming to evaluate direct and indirect benefits of influenza vaccination conducted in 2009–2010[21]. Seven hundred and ninety-six households with at least one child were enrolled, and one child 6–17 years of age in each household was randomly selected to receive either a single dose of trivalent inactivated influenza vaccination or saline placebo. We collected serum samples from every household member at enrollment to the study in August–December 2009 and at the end of the study in August–December 2010. We also collected a third serum sample from all household members of 33% of households in April 2010. Children who received influenza vaccination or placebo also provided an additional serum sample 1 month after vaccination. In total, we

collected up to four sequential serum samples from participants, and the majority of participants provided two serum samples. For children who received vaccine or placebo, the serum samples after vaccination were used as baseline instead of the serum samples collected at enrollment. Infection for an individual was defined by 4-fold or greater rise in at least one paired serum drawn from that individual. We included all households in which at least one household member had paired sera that covered the epidemic in the analyses. Details of sample size justification, Randomization, Allocation Concealment, and Blinding were reported in a previous study[21] and summarized in the Supplementary Methods.

**Ethics**. All participants aged 18 years and older gave written informed consent. Proxy written consent from parents or legal guardians was obtained for participants aged 17 years and younger, with additional written assent from those aged 8 to 17 years. The study protocol was approved by the Institutional Review Board of the University of Hong Kong and by the Hong Kong Department of Health Ethics Committee.

**Laboratory methods**. All serum specimens were tested in parallel for antibody responses to B/Brisbane/60/2008-like (Victoria lineage) by hemagglutination inhibition assays in serial doubling dilutions from an initial dilution of 1:10 using standard methods[27]. Antibody titers were the reciprocal of the highest dilution that completely prevented hemagglutination.

**Model details and inference**. We developed a statistical framework to estimate the probability of getting infected in the community during the epidemic period and the probability of within-household person-to-person transmission from the serologic data. Such inference is challenging because the chains of transmission are unobserved and we only know the final infection status of each individual at the end of the epidemic, also denoted final size data. In such context, methods have been developed since the 1980 s to perform robust parameter inference that equate to integrating the likelihood over all possible chains of transmission consistent with the data[19,39–41]. Here, we used a method based on directed graphs (digraph) described in detail in Cauchemez et al.[19]. and summarized in the Supplementary Methods. In short, a household of size $n$ is represented by a random directed graph with $n$ vertices, each representing a household member. Edges are added to represent possible transmission events. An edge between individual $j$ and individual $i$ indicates that if individual $j$ gets infected, then individual $i$ will get infected too. An edge between the community and individual $i$ indicates that individual $i$ will get infected.

We considered the digraph as augmented data since the chains of transmissions were unobserved. We used a data augmentation Markov chain Monte Carlo approach to jointly explore the parameters and digraph space and estimate the posterior distribution of the model parameters[19,39] (Supplementary Methods).

**Model specification**. We used this statistical framework to evaluate the direct and indirect effect of vaccination that may be obtained by blocking introduction of influenza virus in households, accounting for other possible factors that may affect the transmission dynamics in households. Therefore, age, level of pre-season HAI titers, and direct effects of vaccination were considered as factors potentially affecting transmission. We defined individuals ≤ 18 and > 18 years of age as children and adults, respectively. We defined HAI titers of ≤ 10, between 20–40 and ≥ 80 as low, intermediate and high level of titers, respectively. We considered models that allowed the probability of household transmission to vary with the number of household members, and models that allowed for a difference in infectivity between children and adults.

We also estimated the proportion of cases attributed to household transmission, using the method described by Cauchemez et al.[19]. For each parameter vector drawn from the posterior distribution, we simulated epidemics in households with the household transmission parameters unchanged or being set to zero. The case counts difference between these two scenarios gave this proportion.

**Model adequacy**. We assessed the model adequacy by comparing the observed and expected number of infections in households (Supplementary Methods). A simulation study was conducted to demonstrate that our algorithm could provide unbiased estimates of model parameters (Supplementary Methods).

**Model comparison**. Deviance Information Criterion (DIC) was used for model comparison[42]. Smaller DIC indicates a better model fit. DIC differences > 5 were considered as substantial improvement[43]. Since the likelihood of observed data was not available, DIC cannot be directed evaluated for a given model[44]. Therefore, we used an importance sampling approach to estimate the likelihood for the observed data and evaluate the DIC[19,45] (Supplementary Methods).

**Model prediction**. We conducted a simulation study to evaluate the indirect benefit from two vaccine strategies, (1) vaccinating one child in each household (as in our trial), (2) vaccinating all children in each household. Ten thousand epidemics were simulated in 150,000 households with parameters drawn from their posterior distribution. For each vaccine strategy, we conducted a simulation with

record of the digraphs, so that the source of each infection could be determined (Supplementary Methods). Hence, for a group of individuals, the probability of infection from the community (household) could be estimated by the number of infections from the community (household) in that group over the number of individuals in that group, with very high accuracy due to large number of households. For a small group of infections with inconclusive source (maximum 4% in all simulations), half of them were assigned to be infected from community. Sensitivity analyses were conducted under the assumption that all of them were infected from the community (households). For a given group (children or adults) and a given source of infection (household, community or both), the indirect protection due to a vaccine strategy was measured by the relative probability (in term of ratio) of infection in that group and from that source under this vaccine strategy, compared to the probability of infection under no vaccination strategy (Supplementary Methods).

**Reporting summary**. Further information on experimental design is available in the Nature Research Reporting Summary linked to this article.

**Code availability**. Statistical analyses were conducted using R version 3.2.4 (R Foundation for Statistical Computing, Vienna, Austria). Code is available at Dryad: https://doi.org/10.5061/dryad.cj62621.

## Data availability

The data that were used in this study are available at Dryad: https://doi.org/10.5061/dryad.cj62621.

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

## Acknowledgements

We thank Chan Kit Man, Kwok Hung Chan, Calvin Cheng, Lai-Ming Ho, Ho Yuk Ling, Nicole Huang, Lam Yiu Pong, Tom Lui, Edward Ma, Sophia Ng, Tong Hok Leung, Loretta Mak, Winnie Wai, Jessica Wong, Kevin Yau, and Jenny Yuen for research support. This study was supported by the Research Fund for the Control of Infectious Diseases of the Health, Welfare and Food Bureau of the Hong Kong SAR Government (grant CHP-CE-03), the Theme-based Research Scheme project no. T11-705/14N from the Hong Kong Government, the National Institute of General Medical Sciences (grant U54 GM088558 to the Harvard Center for Communicable Disease Dynamics to B.J.C, MIDAS initiative grant 1U01GM110721-01 to S.C., and U54 GM111274-01 to Center for Statistics and Quantitative Infectious Diseases to T.K.T.), L'Oreal Hong Kong (research scholarship to T.K.T), the Laboratory of Excellence Integrative Biology of Emerging Infectious Diseases (research funding to S.C.), and AXA Research Fund to S.C. The funding bodies had no role in study design, data collection and analysis, preparation of the manuscript, or the decision to publish.

## Author contributions

T.K.T., B.J.C. and S.C. designed this study. V.J.F., D.K.M.I., R.A.P.M.P., H.C.S., G.M.L., J.S.M.P., and B.J.C. collected data. T.K.T. and V.J.F. analyzed data. T.K.T. wrote the first draft of the paper. All authors contributed to critical appraisal of the results and revision of the manuscript. B.J.C. and S.C. contributed equally to this work.

## Additional information

**Competing interests:** B.J.C has received research funding from Sanofi Pasteur. The remaining authors declare no competing interests.

