## [Peer Review File · Nature Communications]

Reviewers' Comments:

Reviewer #1:

Remarks to the Author:

General comments on the paper:

1. Key results The authors provide quantitative estimates of the effect of indirect protection caused by vaccinating one child in a household against influenza. This is achieved by defining a stochastic model of disease spread, and fitting this model to data from a randomized control trial using Bayesian Markov chain Monte Carlo methods.

2. Validity One possible problem is the way that the infection rate for community-acquired infection is defined. The within-household transmission appears to be on a time-scale where one unit equals the duration of time during which an infected individual can infect another household member. It is not clear whether or not the community-acquired infection is defined on the same time-scale or not. If not, then the analysis presented is incorrect. The authors need to address this issue.

3. Originality and significance Estimating the benefits of indirect vaccination is an important contribution with clear consequences for public health policy. The approach taken here is novel and appropriate.

4. Data and methodology The available data are appropriate for the analysis. The only possible problem with the methodology is described in point 2 above. The authors perform model-checking which is a very positive aspect of their analysis.

5. Conclusions The conclusions appear to be valid (if the analysis is correct).

6. Suggested improvements The Supporting Information is unclear in a few places. It should be sufficient for the reader to be able to completely understand what has been done, and replicate the analysis (if the reader also had the same data available).

Specific comments on the paper (line numbers refer to LH margin numbers)

p3, 36 It isn't clear what "household contacts" means. Maybe write "household contacts (i.e. other members of the household)"?

p5, 94 Same comment as above.

p8, 162 "risk of infection from household member" → "risk of infection from a household member"

p8, 164-5 Same comment as above.

p11, 246 "4 time" → "4 times"

p12, 278 "from household" → "from a household"

p14, 318 "that obtained by" → "that may be obtained by"

p14, 327+ In the digraph augmented data method, it looks as if you can directly estimate the proportion of cases attributed to household transmission (since in the MCMC algorithm you keep track of information like the table on page 3 of Supporting Information). Or is this not possible here?

Specific comments on the Supporting Information

p2, 5 “other household member” → “other household members”

p2, 6 “3 round” → “3 rounds”

p2, -11 “We used a proxy...” – this is not very clear. What exactly is being estimated?

p2, -9 “provided” → “that provided”

p3, 1.1 The text after Bayes’ formula is very inaccurate. Specifically, $P(y|G)$ is not “consistence” – presumably it is actually a function which equals 1 if G agrees with y and zero if it does not. Also $P(G|\theta)$ is not “the construction of the digraph” – presumably it is the probability of the digraph G given θ . Finally $P(\theta)$ is presumably the prior density function of θ , not “the distributions of the model parameters”.

p4, 14 The equation for $\lambda^{jk}(\theta)$ has indicator functions missing (it should have $I(hs=4)$ and $I(hs>4)$, specifically).

p4, 16 “were the susceptibility” → “was the susceptibility”

p4 Consider the formula for $P(v^{jk} = 1 | \theta)$. The right-hand side of the formula is the probability that a Poisson process of rate $\lambda^{jk}(\theta)$ has no points in one unit of time. The authors are therefore presumably assuming that each individual who is infected remains infectious for one unit of time. This assumption should be clearly stated. But what is less clear is how this assumption affects the definition of $\lambda^{0k}(\theta)$, since this rate should also be with respect to the same unit of time. If it is not, then the analysis is flawed. It looks as if the authors used weeks as the time unit (text at top of page 5). This might also explain why the authors estimate the proportion of household infection to be so low. It might be that the values used for $\lambda^{0k}(\theta)$ are on the wrong time-scale?

p4 In the main manuscript there is a quantity called “ P ” (p8, line 159). How is P related to the material on this page?

p5, 10 “where β_1 were” → “there β_1 was”

p5, 13 “ β_3 were” → “ β_3 was”

p6, 9, 12, 14 “priors” → “prior”

p6, -13 “followings” → “following”

p6, -12 “missing in” → “missing values in” (?)

p7, 6 “from all the non-edge” → “from all the non-edges”

p7, -3 “edge” → “edges” (twice in this line)

p9, 2 “burin” → “burn-in”

p9 How was DIC calculated?

p11 FigureS1 caption: “infection for child” → “infection for a child”

Reviewer #2:

Remarks to the Author:

In this manuscript, the authors undertake a large study to evaluate the indirect effects of influenza vaccine. In a previously conducted RCT, they followed household contacts of children assigned to either vaccination or placebo over the 2009-2010 influenza season during an influenza B outbreak in Hong Kong and assessed the change in HAI titers among participants and their household contacts by collecting paired sera samples pre and post influenza season. In this secondary analysis, the authors find evidence of limited within-household transmission and minimal indirect protection to household contacts of vaccinated children. They develop statistical and mathematical models to examine the transmission dynamics and they predict that indirect protection would be higher under a scenario where greater transmission occurs within the household and when all children in a household are vaccinated. The manuscript addresses a very interesting topic.

Please clarify in the abstract which participants the analyses of pre-season titers applies to – all contacts from both vaccinated and placebo households? Please specify whether the comparison group for which adults had lower susceptibility to infection is all children or all vaccinated children.

The post-randomization inclusion/exclusion criteria applied to this study/analysis would be clearer if the authors included a CONSORT flow diagram as recommended for the transparent reporting of clinical trials: <http://www.consort-statement.org/consort-statement/flow-diagram>

How did the authors assess whether vaccinees and placebo recipients were infected? This applies to the direct effect of vaccination reported in the abstract and discussion. Since infections are being defined by a 4-fold rise in titers based on paired sera, which timepoints are being used to assess whether vaccinees/placebo recipients were infected and are any considerations made given the ceiling effect of HAI among vaccinees? This raises questions, as noted in the limitations, given that vaccinated individuals can have very high HAI tiers following vaccination.

Figure 1 is helpful for clarifying when sampling “rounds 1-3” took place (dates) relative to the dates of typical influenza transmission in the locations where the participants were recruited. It suggests that comparisons are between sera collected in rounds 1 and 3 capture the majority of influenza transmission since substantial transmission is estimated to be occurring after round 2 collection. Could the authors comment on which data were used from each of the timepoints in the statistical analyses and to inform the models? It is not clear at present.

When evaluating the risk of household infection, the authors claim that the relative risk was nearly halved when comparing the “no vaccination” scenario to the two vaccination strategies they assessed

(page 7, lines 148-151), however the confidence intervals were quite wide (0.07-1.63 in one case) and the precision of those estimates is limited.

The claim in the abstract that 30% of infections in an influenza A outbreak can be attributed to household transmission, compared to the 10% of infections estimated in this study during an influenza B outbreak, is not a result of this study, but rather, it is cited from another study and used as a scenario to evaluate the degree of indirect protection under that specific scenario. This should be clarified in the abstract.

It is important to include the estimates of direct protection in the results section, prior to reporting the infections among child and adult contacts and to clarify the estimates of direct protection as these, too, are noted to be the primary aim of this study in the discussion but the results do not appear in the results section. It is also noted in the discussion that data on PCR-confirmed cases was available from the original trial, so it would be helpful to know the sens/spec of the serological definition of an infection used in this study for those participants for whom both outcomes were evaluated.

The finding that the "...impact [of vaccination] on the overall risk of infection in household contacts was small because household transmission only represented a small proportion of all transmission events" does not seem too surprising. Perhaps the authors could emphasize how their results might translate to recommendations for vaccination strategies? For instance, given low household transmission, it is even more important for individuals to be vaccinated so that they can reduce their risk of infection via direct protection rather than relying on indirect effects of vaccinated household members.

In the discussion about why the relative contribution of household transmission was lower than the authors expected in this study, it would be helpful to include a discussion of the influenza vaccination coverage by age group in the population from which the trial participants were drawn (the authors note it was low/10% overall).

It is not clear that the final statement "vaccinating all children in households may be necessary in order to have indirect protection" follows from the evidence presented. The evidence seems to suggest that even if all children were vaccinated, indirect protection could still be minimal if household transmission was limited.

Reviewer #1 (Remarks to the Author):

General comments on the paper:

1. Key results: The authors provide quantitative estimates of the effect of indirect protection caused by vaccinating one child in a household against influenza. This is achieved by defining a stochastic model of disease spread, and fitting this model to data from a randomized control trial using Bayesian Markov chain Monte Carlo methods.

Response 1: We also thank the reviewers for their very detailed and constructive comments that helped us clarifying the manuscript.

2. Validity: One possible problem is the way that the infection rate for community-acquired infection is defined. The within-household transmission appears to be on a time-scale where one unit equals the duration of time during which an infected individual can infect another household member. It is not clear whether or not the community-acquired infection is defined on the same time-scale or not. If not, then the analysis presented is incorrect. The authors need to address this issue.

Response 2: We appreciate that the description of our model may have lacked clarity in the original manuscript. This description has therefore been largely rewritten in the revised manuscript.

Our analysis relies on a type of inferential method called “final size data analysis” that has been extensively used to study influenza household transmission and is therefore well established. The validity of such approach has been demonstrated with numerous datasets since the 80s (see for example: Longini and Koopman AJE 1982; O’Neill et al JRSS C 2000; Demeris and O’Neill JRSS B 2005; Cauchemez et al PLoS Pathogens 2014).

A strength of the approach is that it only requires the final infection status of household members to be able to characterize transmission (no information required about the timing of infections).

The only time period of interest in such approach is **the period during which the epidemic occurred**. The model aims to predict the number of persons that were infected

in the household during this period accounting for the two possible sources of infection (community-based and within household transmission) as illustrated in the following graph:

To that end, the model assumes that individuals can be infected in the community with a certain probability p_C (probability of community acquired infection) during the epidemic period and that there is also a probability p_H of person-to-person transmission in the household. Since timings of infection are unobserved, these approaches equate to integrating over all the possible scenarios, so that there is no issue with time scales. Additional theoretical justifications concerning the validity of the approach can be found in theoretical journals such as *Biometrics* (Longini et al. 1982) or *JRSS B* (Demeris et al. 2005).

We appreciate that the mention in the Supplement of a risk of infection in the community that varied weekly may have caused rightly confusion. In retrospect, we believe that this was a mistake. Indeed, in the original submission, for a subset of cases, we used information on whether infection occurred between rounds 1-2 or 2-3 to inform the value of the community risk. However, this clearly violated the premise of these “final size approaches” that should only rely on the final infection status to infer parameters. In the revised manuscript, this error was corrected. The probability of getting infected in the community during the epidemic now only depends on individual characteristics. The confusing section in the Supplement about weekly risks of community-acquired infection

varying with syndromic surveillance data has been removed. Again, we thank the referee for this important comment.

In the revised manuscript, we have clarified the description of the model and the theoretical justification for our approach:

***“Bayesian inference.** We developed a statistical framework to estimate the probability of getting infected in the community during the epidemic period and the probability of within-household person-to-person transmission from the serologic data. Such inference is challenging because the chains of transmission are unobserved and we only know the final infection status of each individual at the end of the epidemic, also denoted “final size” data. In such context, methods have been developed since the 80s to perform robust parameter inference that equate to integrating the likelihood over all possible chains of transmission consistent with the data¹⁻⁴. Here, we used a method based on directed graphs (digraph) described in detail in Cauchemez et al¹. In short, a household of size n is represented by a random directed graph with n vertices, each representing a household member. Edges are added to represent possible transmission events. An edge between subject j and subject i indicates that if subject j gets infected, then subject i will get infected too. An edge between the community and subject i indicates that subject i will get infected.*

We considered the digraph as augmented data since the chains of transmissions were unobserved. We used a data augmentation Markov chain Monte Carlo approach to jointly explore the parameters and digraph space and estimate the posterior distribution of the model parameters^{1,2} (SI appendix, Section 2.2).”

3. Originality and significance: Estimating the benefits of indirect vaccination is an important contribution with clear consequences for public health policy. The approach taken here is novel and appropriate.

Response 3: We thanks the reviewer for positive comment for our work.

4. Data and methodology: The available data are appropriate for the analysis. The

only possible problem with the methodology is described in point 2 above. The authors perform model-checking which is a very positive aspect of their analysis.

Response 4: We have clarified this in point 2 above. We thank the reviewer for appreciation for the model-checking.

5. Conclusions: The conclusions appear to be valid (if the analysis is correct).

Response 5: We have clarified this in point 2 above and make sure our conclusions were supported by appropriate evidence.

6. Suggested improvements The Supporting Information is unclear in a few places. It should be sufficient for the reader to be able to completely understand what has been done, and replicate the analysis (if the reader also had the same data available).

Response 6: We have modified the presentation of the modelling and the assumption for the risk of infection from community, and also added relevant references for the approach, so that reader could completely understand what we have done and perform replication.

Specific comments on the paper (line numbers refer to LH margin numbers)

p3, 36 It isn't clear what "household contacts" means. Maybe write "household contacts (i.e. other members of the household)"?

Response 7: We agree and modified accordingly.

p5,94 Same comment as above.

Response 8: Modified accordingly.

P8, 162 "risk of infection from household member" → "risk of infection from a household member"

Response 9: Modified accordingly.

P8, 164-5 Same comment as above.

Response 10: Modified accordingly.

P11, 246 “4 time” → “4 times”

Response 11: Modified accordingly.

P12, 278 “from household” → “from a household”

Response 12: Modified accordingly.

P14, 318 “that obtained by” → “that may be obtained by”

Response 13: Modified accordingly.

p14, 327+ In the digraph augmented data method, it looks as if you can directly estimate the proportion of cases attributed to household transmission (since in the MCMC algorithm you keep track of information like the table on page 3 of Supporting Information). Or is this not possible here?

Response 14: The reviewer is right that we can keep track of information like the table on page 3 of supporting information. However, the source of infection may not be unique, and hence it could be difficult to define if an individual was infected from a household member or from the community. Consider the following example:

	Participant 1	Participant 2
Community	1	1
Participant 1	0	1
Participant 2	0	0

In this digraph, participant 2 could have been infected either in the community or by participant 1. Therefore, we used the approach in Cauchemez et al. to estimate this proportion¹.

Specific comments on the Supporting Information

P2, 5 “other household member” → “other household members”

Response 15: Modified accordingly.

P2, 6 “3 round” → “3 rounds”

Response 16: Modified accordingly.

P2, -11 “We used a proxy...” – this is not very clear. What exactly is being estimated?

Response 17: Thank you. We agree that this part was unclear. As indicated in response 2, in the original submission the risk of community infection in the original analysis depended on partial data about the timing of infection, but this violated the premise of “final size inference”. In the revised manuscript, we assume that the risk of infection in the community during the whole epidemic only varies with individual characteristics. We therefore no longer rely on data from syndromic surveillance.

P2, -9 “provided” → “that provided”

Response 18: Modified accordingly.

P3, 1.1 The text after Bayes’ formula is very inaccurate. Specifically, $P(y|G)$ is not “consistence” – presumably it is actually a function which equals 1 if G agrees with y and zero if it does not. Also $P(G | \theta)$ is not “the construction of the digraph” – presumably it is the probability of the digraph G given θ . Finally $P(\theta)$ is presumably the prior density function of θ , not “the distributions of the model parameters”.

Response 19: Thank you for spotting these mistakes. We have modified the corresponding text (underline indicates modification):

“Denote G the digraph representing the potential transmission chain in households, y the observed data, and θ the parameter vector:

$$P(G, \theta | y) \propto P(y|G)P(G|\theta)P(\theta)$$

Here, $P(y|G)$ is an indicator function equal to 1 if the infection status of all participants derived from the digraph G agrees with the observed infection status y . $P(G|\theta)$ is the

probability of digraph G given the parameters θ . $P(\theta)$ is the prior density function of the model parameters θ ."

P4, 14 The equation for $\lambda^{jk}(\theta)$ has indicator functions missing (it should have $I(hs=4)$ and $I(hs>4)$, specifically).

Response 20: Modified accordingly.

P4, 16 "were the susceptibility" → "was the susceptibility"

Response 21: Modified accordingly.

P4 Consider the formula for $P(v^{jk} = 1 \mid \theta)$. The right-hand side of the formula is the probability that a Poisson process of rate $\lambda^{jk}(\theta)$ has no points in one unit of time. The authors are therefore presumably assuming that each individual who is infected remains infectious for one unit of time. This assumption should be clearly stated.

But what is less clear is how this assumption affects the definition of $\lambda^{\{0k\}}(\theta)$, since this rate should also be with respect to the same unit of time. If it is not, then the analysis is flawed. It looks as if the authors used weeks as the time unit (text at top of page 5). This might also explain why the authors estimate the proportion of household infection to be so low. It might be that the values used for $\lambda^{\{0k\}}(\theta)$ are on the wrong time-scale?

Response 22: See response 2. We appreciate that the description of our model in the original submission lacked clarity. As it should now be clear from response 2, the unit of time we consider is the full duration of the epidemic and we model the final infection status of each individual at the end of the epidemic. We apologize for the misleading mention of community risks that varied weekly in the original manuscript; this error has been corrected in the revised manuscript.

P4 In the main manuscript there is a quantity called "P" (p8, line 159). How is P related to the material on this page?

Response 23: It is the proportion of cases that were attributed to household transmission.
We have clarified this in the main text.

P5, 10 “where beta_1 were” → “there beta_1 was”

Response 24: Modified accordingly.

P5, 13 “beta_3 were” → “beta_3 was”

Response 25: Modified accordingly.

P6, 9,12 ,14 “priors” → “prior”

Response 26: Modified accordingly.

P6, -13 “followings” → “following”

Response 27: Modified accordingly.

P6, -12 “missing in” → “missing values in” (?)

Response 28: Modified accordingly.

P7, 6 “from all the non-edge” → “from all the non-edges”

Response 29: Modified accordingly.

P7, -3 “edge” → “edges” (twice in this line)

Response 30: Modified accordingly.

P9, 2 “burin” → “burn-in”

Response 31: Modified accordingly.

P9 How was DIC calculated?

Response 32: We have added the formula of DIC in section 4 in the appendix.

P11 FigureS1 caption: “infection for child” → “infection for a child”

Response 33: Modified accordingly.

Reviewer #2 (Remarks to the Author):

In this manuscript, the authors undertake a large study to evaluate the indirect effects of influenza vaccine. In a previously conducted RCT, they followed household contacts of children assigned to either vaccination or placebo over the 2009-2010 influenza season during an influenza B outbreak in Hong Kong and assessed the change in HAI titers among participants and their household contacts by collecting paired sera samples pre and post influenza season. In this secondary analysis, the authors find evidence of limited within-household transmission and minimal indirect protection to household contacts of vaccinated children. They develop statistical and mathematical models to examine the transmission dynamics and they predict that indirect protection would be higher under a scenario where greater transmission occurs within the household and when all children in a household are vaccinated. The manuscript addresses a very interesting topic.

Response 34: We thanks the reviewer for their supportive and constructive comments.

Please clarify in the abstract which participants the analyses of pre-season titers applies to – all contacts from both vaccinated and placebo households? Please specify whether the comparison group for which adults had lower susceptibility to infection is all children or all vaccinated children.

Response 35: The estimation of protection from pre-season titers were based on all unvaccinated households contacts from all households. Unvaccinated adults had lower susceptibility to infection when compared with unvaccinated children. We have clarified this in the main text (underline indicates modification):

“Based on the data from both vaccinated and control households, we estimated that unvaccinated adult contacts had lower susceptibility than unvaccinated child contacts

(relative susceptibility: 0.39; 95% CI: 0.28, 0.54, Figure 2A). Ignoring this difference substantially worsened model fit (Δ DIC: 33.0). We also estimated that the relative susceptibility of unvaccinated contacts with an intermediate level of HAI titer and with a high level of HAI titer was 0.48 (95% CI: 0.23, 0.90) and 0.42 (95% CI: 0.17, 0.89), respectively, compared with those with a low level of HAI titer respectively (Figure 2B). The model without protection effect from pre-season HAI titers performed substantially worse (Δ DIC: 14.0)."

We no longer present this results in the abstract, since we had to shorten it to present the main results – the direct and indirect effect of vaccination, as the word limit is 150.

The post-randomization inclusion/exclusion criteria applied to this study/analysis would be clearer if the authors included a CONSORT flow diagram as recommended for the transparent reporting of clinical trials: <http://www.consort-statement.org/consort-statement/flow-diagram>

Response 36: Thank you for the suggestion. We have added a flow diagram in the manuscript.

How did the authors assess whether vaccinees and placebo recipients were infected? This applies to the direct effect of vaccination reported in the abstract and discussion. Since infections are being defined by a 4-fold rise in titers based on paired sera, which timepoints are being used to assess whether vaccinees/placebo recipients were infected and are any considerations made given the ceiling effect of HAI among vaccinees? This raises questions, as noted in the limitations, given that vaccinated individuals can have very high HAI tiers following vaccination.

Response 37: Children who received influenza vaccination or placebo also provided an additional serum sample 1 month after vaccination. Therefore, for children who received vaccine or placebo, the serum samples after vaccination were used as baseline instead of the serum samples collected at enrolment. We have added the following in the study design in method section:

"For children who received vaccine or placebo, the serum samples after vaccination were used as baseline instead of the serum samples collected at enrolment."

While there could be ceiling effect among vaccinees, this effect should be limited since in our study only 23/467 (5%) of vaccinees had an HAI titer of >640, while our ceiling of HAI titer is 2560. Also, the estimate of direct vaccine efficacy from our analysis was 73%, which was similar to the previous estimate 66% based on PCR⁵. Therefore, we did not make any adjustment on this. We have clarified this in the limitations in the discussion:

“However, the estimate of vaccine efficacy based on HAI titers here was very similar to that based on PCR-confirmed influenza¹⁹. Moreover, only 5% (23/467) of vaccinees had a post-vaccination HAI titers of >640, while our ceiling for HAI titers was 2560.”

Figure 1 is helpful for clarifying when sampling “rounds 1-3” took place (dates) relative to the dates of typical influenza transmission in the locations where the participants were recruited. It suggests that comparisons are between sera collected in rounds 1 and 3 capture the majority of influenza transmission since substantial transmission is estimated to be occurring after round 2 collection. Could the authors comment on which data were used from each of the timepoints in the statistical analyses and to inform the models? It is not clear at present.

Response 38: We thanks for the reviewer for this important comment. In our analysis, infection was defined as 4-fold or greater rise in at least one paired serum. If we denote AT^1, AT^2, AT^3 the HAI titer levels for the serum drawn at round 1, 2 and 3 respectively,

infection was therefore defined as $\frac{AT_{ij}^3}{AT_{ij}^1} \geq 4$ when AT_{ij}^2 was not available, and as $\frac{AT_{ij}^2}{AT_{ij}^1} \geq$

4 or $\frac{AT_{ij}^3}{AT_{ij}^2} \geq 4$ when AT_{ij}^2 was available. We added the following in the revised manuscript:

In the study design in method section:

“Infection was defined by 4-fold or greater rise in at least one paired serum drawn from that individual.”

In appendix:

“Infection was defined as a 4-fold or greater rise in at least one paired sera. and therefore we defined a variable y_{ij} , where $y_{ij} = 1$ if $\frac{AT_{ij}^3}{AT_{ij}^1} \geq 4$ when AT_{ij}^2 was not available, and $y_{ij} = 1$ if

$\frac{AT_{ij}^2}{AT_{ij}^1} \geq 4$ or $\frac{AT_{ij}^3}{AT_{ij}^2} \geq 4$ when AT_{ij}^2 was available.”

When evaluating the risk of household infection, the authors claim that the relative risk was nearly halved when comparing the “no vaccination” scenario to the two vaccination strategies they assessed (page 7, lines 148-151), however the confidence intervals were quite wide (0.07-1.63 in one case) and the precision of those estimates is limited.

Response 39: The derivation of the intervals was inappropriate in the original manuscript. This has been corrected in the revisions. We now use the following approach to derive posterior predictive intervals:

For each parameter vector from the posterior distribution, we simulated influenza outbreaks in a large number of households (150,000). We repeated this for 10000 parameter vectors drawn from the posterior distribution to get the 95% posterior predictive intervals. This approach ensures that we correctly captures the effect of parameter uncertainty on model predictions.

We added the following:

In the model prediction in method section:

“10,000 epidemics were simulated in 150000 households with parameters drawn from their posterior distribution.”

In the model prediction in section 5 in Appendix:

“We simulated 10,000 epidemics in 150000 households with parameters drawn from their posterior distribution. The structure of a simulated household was identical to that of an household randomly drawn in the study. With this approach, we were able to derive 95% posterior predictive intervals that correctly captured the effect of parameter uncertainty on model predictions.”

The claim in the abstract that 30% of infections in an influenza A outbreak can be

attributed to household transmission, compared to the 10% of infections estimated in this study during an influenza B outbreak, is not a result of this study, but rather, it is cited from another study and used as a scenario to evaluate the degree of indirect protection under that specific scenario. This should be clarified in the abstract.

Response 40: We have removed those sentences from the abstract because of word limits. We have checked our manuscript to make sure that readers are informed that the 30% of infections in an influenza A outbreak is from other studies. We have clarified this in the discussion:

“The estimated proportion of household transmission was surprisingly low given other studies found it to be closer to 30%^{19,20} for influenza A epidemic.”

It is important to include the estimates of direct protection in the results section, prior to reporting the infections among child and adult contacts and to clarify the estimates of direct protection as these, too, are noted to be the primary aim of this study in the discussion but the results do not appear in the results section. It is also noted in the discussion that data on PCR-confirmed cases was available from the original trial, so it would be helpful to know the sens/spec of the serological definition of an infection used in this study for those participants for whom both outcomes were evaluated.

Response 41: We agree that direct protection was one of the primary aim. We now report the vaccine efficacy, computed by one minus relative susceptibility, in a new section:

*“**Direct effect of vaccination:** We found that children who had received influenza vaccination had a lower susceptibility compared with children who had received the placebo (relative susceptibility: 0.29; 95% CI: 0.17, 0.47, Figure 2A). Models assuming no direct effect of vaccination performed substantially worse (ΔDIC : 32.4). The vaccine efficacy, computed by one minus relative susceptibility, was therefore 71% (95% CI: 53%, 83%).”*

We decided to define infections based on serology, due to the fact that PCR was unable to detect asymptomatic infection, because swabs were only collected when the participants reported symptoms. We have added this in the discussion:

“We decided to use serology instead of PCR because PCR was unable to detect asymptomatic

infection since swabs were only collected when the participants reported symptoms.”

The finding that the “...impact [of vaccination] on the overall risk of infection in household contacts was small because household transmission only represented a small proportion of all transmission events” does not seem too surprising. Perhaps the authors could emphasize how their results might translate to recommendations for vaccination strategies? For instance, given low household transmission, it is even more important for individuals to be vaccinated so that they can reduce their risk of infection via direct protection rather than relying on indirect effects of vaccinated household members.

Response 42: Thanks for the suggestion. We agree that our results suggested that the indirect protection is limited (20%), compared with the direct protection (73%). Therefore, we agree that our results suggested that the direct effect of vaccination is important given that indirect protection from household members is limited. We have added this in the discussion section.

In the discussion about why the relative contribution of household transmission was lower than the authors expected in this study, it would be helpful to include a discussion of the influenza vaccination coverage by age group in the population from which the trial participants were drawn (the authors note it was low/10% overall).

Response 43: Thank you for the suggestion, we agree this would be helpful. In our study, 48/534 (9%) children and 177/1700 (10%) adults were vaccinated. However, this may not reflect the population coverage as our studies selected household with at least one unvaccinated child. From a household transmission study in Hong Kong^{6,7}, 39/218 (18%) of child contacts of index cases and 111/923 (12%) of adult contacts of index cases were vaccinated. Another household study reported that the vaccination coverage for elderly was 27%⁸. We have added this result in the discussion:

“If a substantial proportion of individuals was vaccinated in the community, the risk of infection from the community could decrease due to herd immunity as shown in other studies¹³⁻¹⁵. In our study, 48/534 (9%) children and 177/1700 (10%) adults were vaccinated. However, this may not reflect the population coverage as our study selected household with at

least one unvaccinated children. From a separated household transmission study conducted in Hong Kong^{23,24}, 39/218 (18%) of child contacts and 111/923 (12%) of adult contacts of index cases were vaccinated. Another household study reported that the vaccine coverage for elderly was 27%²⁵.”

It is not clear that the final statement “vaccinating all children in households may be necessary in order to have indirect protection” follows from the evidence presented. The evidence seems to suggest that even if all children were vaccinated, indirect protection would still be minimal if household transmission was limited.

Response 44: We agree this would be a better interpretation of our results. We have modified our final statement to *“We have tested the degree of indirect protection in different scenarios, and found that vaccinating all children in a household provided limited indirect protection (20%), which is lower than the direct vaccine efficacy (73%). This suggests that the benefits of individual vaccination remain important even when other household members are vaccinated.”*

Reviewers' Comments:

Reviewer #1:

Remarks to the Author:

NCOMMS-18-10788A:

"Are parents and siblings indirectly protected when a child is vaccinated against influenza? by Tsang et al.

The manuscript is now largely fine, but one potential issue (from my original report) is not completely resolved. The original comment, on the Supporting Information (sections 1.2.1 and 1.2.2 in the revised version) was:

"Consider the formula for $P(v^{\{jk\}} = 1 \mid \theta)$. The right-hand side of the formula is the probability that a Poisson process of rate $\lambda^{\{jk\}}(\theta)$ has no points in one unit of time. The authors are therefore presumably assuming that each individual who is infected remains infectious for one unit of time. This assumption should be clearly stated.

But what is less clear is how this assumption affects the definition of $\lambda^{\{0k\}}(\theta)$, since this rate should also be with respect to the same unit of time. If it is not, then the analysis is flawed. It looks as if the authors used weeks as the time unit (text at top of page 5). This might also explain why the authors estimate the proportion of household infection to be so low. It might be that the values used for $\lambda^{\{0k\}}(\theta)$ are on the wrong time-scale?"

The authors' response makes it clear that there were some problems with the original manuscript and that the authors realise that the time period during which the epidemic occurs is a suitable (implicit) time unit.

However, the points I made still need to be clarified. In both the original and the revised version, both the within-household AND the community-acquired infection models are of the form

$$\text{Probability} = 1 - \exp \{- \lambda(\theta) \}$$

and in both cases (i) the λ is described as a transmission rate (or force of infection, which I assume mean the same thing), so (ii) the probability is the probability of a potential contact occurring during one time unit.

For the community-acquired infection, this time unit is (most naturally) the time period during which the epidemic occurs, which is several months. But this time unit is completely inappropriate for within-household transmission, since there individuals are only likely to be infectious for a period of a few days. In other words, if the time unit is "time for the entire epidemic", then the within-household model should (or could) be

$$\text{Probability} = 1 - \exp \{- \lambda(\theta) T \}$$

where T is some assumed fixed value (e.g. 4 days). Now, with such a model one could then simply redefine $\lambda(\theta) T$ as a new parameter, say $\lambda^*(\theta) = \lambda(\theta) T$, in which case the model is equivalent as the one presented by the authors, but where the model parameter λ^* now has a different interpretation to λ – it is no longer a transmission rate.

At the very least, this all needs to be clarified. Whether or not it affects the results depends crucially

on what the authors are using to obtain their figures and estimates. If all the results are in terms of probabilities, i.e. comparing $1 - \exp(-\lambda)$ for different λ s corresponding to different types of individual etc, then the results should be fine as they are. However, if any of the results are in terms of λ s (e.g. comparing the household and community λ s) then this does not appear to be correct, unless the authors are happy to assume that each person infected in a household is infective for several months.

Reviewers' comments:

Reviewer #1 (Remarks to the Author):

NCOMMS-18-10788A:

“Are parents and siblings indirectly protected when a child is vaccinated against influenza? by Tsang et al.

The manuscript is now largely fine, but one potential issue (from my original report) is not completely resolved. The original comment, on the Supporting Information (sections 1.2.1 and 1.2.2 in the revised version) was:

“Consider the formula for $P(v^{\{jk\}} = 1 \mid \theta)$. The right-hand side of the formula is the probability that a Poisson process of rate $\lambda^{\{jk\}}(\theta)$ has no points in one unit of time. The authors are therefore presumably assuming that each individual who is infected remains infectious for one unit of time. This assumption should be clearly stated.

But what is less clear is how this assumption affects the definition of $\lambda^{\{0k\}}(\theta)$, since this rate should also be with respect to the same unit of time. If it is not, then the analysis is flawed. It looks as if the authors used weeks as the time unit (text at top of page 5). This might also explain why the authors estimate the proportion of household infection to be so low. It might be that the values used for $\lambda^{\{0k\}}(\theta)$ are on the wrong time-scale?”

The authors’ response makes it clear that there were some problems with the original manuscript and that the authors realise that the time period during which the epidemic occurs is a suitable (implicit) time unit.

However, the points I made still need to be clarified. In both the original and the revised version, both the within-household AND the community-acquired infection models are of the form

$$\text{Probability} = 1 - \exp \{- \lambda(\theta) \}$$

and in both cases (i) the λ is described as a transmission rate (or force of infection, which I assume mean the same thing), so (ii) the probability is the probability of a potential contact occurring during one time unit.

For the community-acquired infection, this time unit is (most naturally) the time period during which the epidemic occurs, which is several months. But this time unit is completely inappropriate for within-household transmission, since there individuals are only likely to be infectious for a period of a few days. In other words, if the time unit is “time for the entire epidemic”, then the within-household model should (or could) be

Probability = $1 - \exp \{- \text{lambda} (\text{theta}) T \}$

where T is some assumed fixed value (e.g. 4 days). Now, with such a model one could then simply redefine lambda(theta) T as a new parameter, say lambda*(theta) = lambda (theta) T, in which case the model is equivalent as the one presented by the authors, but where the model parameter lambda* now has a different interpretation to lambda – it is no longer a transmission rate.

At the very least, this all needs to be clarified. Whether or not it affects the results depends crucially on what the authors are using to obtain their figures and estimates. If all the results are in terms of probabilities, i.e. comparing $1 - \exp(\text{lambda})$ for different lambdas corresponding to different types of individual etc, then the results should be fine as they are. However, if any of the results are in terms of lambdas (e.g. comparing the household and community lambdas) then this does not appear to be correct, unless the authors are happy to assume that each person infected in a household is infective for several months.

Response 2. We thank the referee for this important comment. The referee is right that parameter lambda is not a transmission rate and we apologize for leaving this terminology in our first revision. As alluded to by the referee, parameter lambda is just used to define a parametric model for the probability of transmission during the study period, i.e. $P=1-\exp(-\text{lambda})$. In practice, we are just comparing these probabilities of transmission and therefore we agree with the referee that results are fine as they are. We are also not attempting to compare lambdas from the community and the household. In the revised manuscript, we have ensured that we never use the term “rate” or “force of infection” to define lambdas.

For SI Section 1.2.1, it is revised as follows:

“Variable v^{jk} indicates the presence of an edge from individual j to individual k , occurring with a probability:

$$P(v^{jk} = 1|\theta) = 1 - \exp(-\lambda^{jk}(\theta))$$

The formulation of $\lambda^{jk}(\theta)$ is as follows:

$$\lambda^{jk}(\theta) = \{\lambda_{h1}I(hs < 4) + \lambda_{h2}I(hs \geq 4)\} * S_k(\theta),$$

where $\lambda_{h1}, \lambda_{h2}$ are parameters that measure the strength of transmission in households of size < 4 and ≥ 4 , respectively, and $S_k(\theta)$ is the susceptibility component for individual k described in Section 1.2.3.

Reviewers' Comments:

Reviewer #1:

Remarks to the Author:

The revised manuscript addresses the remaining concerns from the reviewers.